# Chandrasekhar’s Conditions for the Equilibrium and Stability of Stars in a Universal Three-Parameter Non-Maxwell Distribution

**DOI:** 10.3390/e27050470

**Published:** 2025-04-26

**Authors:** Wei Hu, Jiulin Du

**Affiliations:** Department of Physics, School of Science, Tianjin University, Tianjin 300072, China; jldu@tju.edu.cn

**Keywords:** complex plasma systems, equilibrium and stability, non-Maxwell distribution, gravity, radiation pressure

## Abstract

The idea of Chandrasekhar’s conditions for the equilibrium and stability of stars is revisited with a new universal three-parameter non-Maxwell distribution. We derive the maximum radiation pressures in the non-Maxwell distribution for a gas star and a centrally condensed star, and thus, we generalize Chandrasekhar’s conditions in a Maxwellian sense. By numerical analyses, we find that the non-Maxwellian distribution usually reduces the maximum radiation pressures in both gas stars and centrally condensed stars as compared with cases where the gas is assumed to have a Maxwellian distribution.

## 1. Introduction

Early in 1916, Eddington noticed the important effect of radiation pressure on stars [1] and showed that radiation pressure works with internal thermal pressure against gravity to prevent stars from collapsing. He mentioned an extraordinary fact that the masses of most stars observed are close to the mass of the Sun. According to his calculations [2], if the mass of a star is less than 10^33^ g, the radiation pressure will be close to 0. In contrast, if the mass of a star is greater than 10^35^ g, the fraction of the gas pressure will be negligible compared to the radiation pressure. The majority of masses of known stars are between 10^33^ and 10^34^ g, where radiation pressure competes with gas pressure. Chandrasekhar’s work may offer an explanation for this factor. In 1936, Chandrasekhar presented the famous inequality for the stable existence of a star in hydrostatic equilibrium [3]:(1)$$\frac{1}{2}G{\left(\frac{4\pi }{3}\right)}^{\frac{1}{3}}{\bar{\rho }}^{\frac{4}{3}}M^{\frac{2}{3}}\leq P_{c}\leq \frac{1}{2}G{\left(\frac{4\pi }{3}\right)}^{\frac{1}{3}}{\rho_{c}}^{\frac{4}{3}}M^{\frac{2}{3}}\;,$$
where *G* is the gravitational constant, *P_C_* is the pressure at the center of a star of mass *M*, $\bar{\rho }$ is the mean density of the star, and *ρ_C_* denotes the density at the center. This is a necessary condition for a star’s stability and describes the equilibrium between its internal pressure and gravitation. From the right-hand side of this inequality, one can obtain the relationship between the proportion of the gas pressure $\beta_{C}$ and the radiation pressure 1 − $\beta_{C}$ at the center of a stable star [4] as follows:(2)$$5.48M_{⊙}{\left(\frac{1-\beta_{c}}{{\beta_{c}}^{4}}\right)}^{\frac{1}{2}}\leq \mu^{2}M,$$
where $M_{⊙}$ is a constant equal to the mass of the sun, *M* is the mass of the star, and *μ* is the mean molecular weight. If the upper limit of 1 − $\beta_{C}$ is written as 1 − $\beta_{*}$, namely,(3)$$1-\beta_{c}\leq 1-\beta_{*}.$$Then the value of 1 − $\beta_{*}$ can be obtained from the equation,(4)$$5.48M_{⊙}{\left(\frac{1-\beta_{*}}{{\beta_{*}}^{4}}\right)}^{\frac{1}{2}}=\mu^{2}M.$$
The upper limit of radiation pressure 1 − $\beta_{*}$ can be determined if the mass *M* of the star and the mean molecular weight *μ* are given. As explained in the paper by Chandrasekhar (1984), such a result follows naturally from the kinetic theory in the Maxwellian sense, where the matter is assumed to satisfy the equation of state of an ideal gas [4]. In addition, Chandrasekhar also showed the effect of radiation pressure on centrally condensed stars. By comparing the electron density in the ideal gas envelope and the degenerate zone, the condition for a star to develop degeneracy can be obtained [5] by(5)1−β<0.0921.
This result has important implications for our understanding of the evolution of stars. A link between the radiation pressure and the mass of a star can be established with Eddington’s standard model, from which the maximum mass of a star with a degenerate core can be estimated. It is worth noting that such results are also obtained from the kinetic theory by assuming that the equation of state of an ideal gas in the Maxwellian sense is satisfied.

On the other hand, many studies have shown that the traditional Maxwellian distribution may not be sufficient to describe the statistical behavior of superheated particles in space and astrophysical plasmas, and a lot of non-Maxwellian distributions have been found. In 1936, Vasyliunas proposed the kappa distribution as a generalization of the Maxwellian distribution based on the observations of OGO1 and OGO3 [6]. In 1995, Cairns et al. proposed a non-Maxwell distribution to explain the density depletions of solitary electrostatic structures observed by the Freja satellite [7]. Based on the kappa distribution, Ali et al. proposed a two-parameter (*r*, *q*)-distribution to study dust plasmas [8]. In 2015, Abid et al. combined the kappa distribution function with the Cairns distribution function and proposed a Vasyliunas–Cairns distribution [9]. These non-Maxwellian distributions have been widely used in various complex space and astrophysical plasmas, such as solar wind [10,11], dusty plasma [12,13], planetary magnetospheres [14,15], and so on. In 2017, Abid et al. introduced a universal three-parameter non-Maxwell distribution function, which contains all of the above non-Maxwellian distributions [16]. The three-parameter distribution function can be written for the *j*th component [17] by(6)$$f_{j}\left(v_{j}\right)=Y_{j}\left(1+\alpha \frac{{v_{j}}^{4}}{{v_{Tj}}^{4}}\right){\left[1+\frac{1}{q-1}{\left(\frac{{v_{j}}^{2}}{X_{r,q}{v_{Tj}}^{2}}\right)}^{r+1}\right]}^{-q},$$
where *Y_j_* is the normalization constant with(7)$$Y_{j}=\left(\frac{3N_{j}}{4\pi {v_{Tj}}^{3}}\right)\frac{\rho_{\alpha ,r,q}}{{X_{r,q}}^{\frac{3}{2}}},$$(8)$$\rho_{\alpha ,r,q}=\frac{\Gamma \left(q\right)}{\left(1+9\alpha \eta_{r,q}\right){\left(q-1\right)}^{\frac{3}{2r+2}}\Gamma \left(\frac{3}{2r+2}+1\right)\Gamma \left(q-\frac{3}{2r+2}\right)},$$(9)$$\eta_{r,q}=\frac{\Gamma \left(q-\frac{3}{2r+2}\right)\Gamma \left(\frac{3}{2r+2}\right)\Gamma \left(q-\frac{7}{2r+2}\right)\Gamma \left(\frac{7}{2r+2}\right)}{\Gamma^{2}\left(q-\frac{5}{2r+2}\right)\Gamma^{2}\left(\frac{5}{2r+2}\right)},$$(10)$$X_{r,q}=\frac{3\Gamma \left(q-\frac{3}{2r+2}\right)\Gamma \left(\frac{3}{2r+2}\right)}{{\left(q-1\right)}^{\frac{1}{r+1}}\Gamma \left(q-\frac{5}{2r+2}\right)\Gamma \left(\frac{5}{2r+2}\right)}.$$In the velocity distribution function (Equation (6)), *v_Tj_* is the thermal speed $\sqrt{kT_{j}/m_{j}}$ for the *j*th component. *q*, *α,* and *r* are three parameters describing the deviations from this distribution function to the Maxwellian one, and they need to satisfy *q* > 1, *α* ≥ 0, and *q*(*r* + 1) > 9/2.

Equation (6) is a universal three-parameter non-Maxwell distribution function, which contains various known non-Maxwell distributions. It is easy to verify as follows: if one takes *α* = 0, the three-parameter distribution can reduce to the (*r*, *q*)-distribution [8]: (11)$$f_{j}(v_{j})∼{\left[1+\frac{1}{q-1}{\left(\frac{v_{j}^{2}}{v_{j,r,q}^{2}}\right)}^{r+1}\right]}^{-q},$$
with$$v_{j,r,q}=\sqrt{\frac{3\Gamma (q-\frac{3}{2r+2})\Gamma (\frac{3}{2r+2})}{{\left(q-1\right)}^{\frac{1}{r+1}}\Gamma (q-\frac{5}{2r+2})\Gamma (\frac{5}{2r+2})}\frac{kT_{j}}{m_{j}}},$$
and if one takes *r =* 0, *α* = 0, and *q* = *κ* + 1, it can reduce to the kappa distribution [18]:(12)$$f_{j}(v_{j})∼{\left(1+\frac{v_{j}^{2}}{\kappa v_{j,\kappa }^{2}}\right)}^{-(\kappa +1)},$$
where the characteristic velocity $v_{j,\kappa }=\sqrt{2kT_{j}(\kappa -3/2)/(\kappa m_{j})}$. When one takes *r* = 0 and *q* = *κ* + 1, it reduces to the Vasyliunas–Cairns distribution [9]:(13)$$f_{j}(v_{j})∼\left(1+\alpha \frac{v_{j}^{4}}{v_{T_{j}}^{4}}\right){\left[1+\frac{v_{j}^{2}}{(2\kappa -3)v_{T_{j}}^{2}}\right]}^{-(\kappa +1)},$$
and when one takes *r* = 0 and *q* → ∞, it further reduces to the Cairns distribution [7]:(14)$$f_{j}(v_{j})∼\left(1+\alpha \frac{v_{j}^{4}}{v_{T_{j}}^{4}}\right)e^{-\frac{v_{j}^{2}}{2v_{T_{j}}^{2}}},$$
with $v_{T_{j}}=\sqrt{kT_{j}/m_{j}}$. Finally, the Maxwellian distribution can be retained when we take *r =* 0, *α* = 0, and *q →* ∞. The Maxwellian distribution is best applicable to systems in a thermal equilibrium state, but the three-parameter distribution may be a more generalized tool to study complex systems in space and astrophysical plasmas with non-Maxwell velocity distributions.

In previous theories about the internal structure of stars, a prevalent view is to consider the highly ionized atoms in the interior of stars as an ideal gas cloud in the Maxwellian sense. However, in fact, the interior of a star and the related astrophysical systems are complex nonequilibrium systems. The recent observations of complex systems in space and astrophysical plasmas may indicate the universal existence of non-Maxwell velocity distributions. There are numerous observations that indicate the interiors of stars and the gas layers of white dwarfs to be complex plasma systems far away from a thermal equilibrium state [19,20,21,22]. There is also experimental evidence suggesting the widespread existence of non-Maxwellian velocity distributions in stellar interiors and astrophysical environments [23,24,25,26,27]. Chandrasekhar’s condition of the equilibrium and stability for a star was generalized in the nonextensive kinetic theory [28], where the gas is assumed to be the power-law *q*-distribution in nonextensive statistics, and now, we know that the *q*-distribution is equivalent to the kappa distribution in space and astrophysical plasmas [29].

Due to the extremely high temperatures of stars and white dwarfs, the ionization of their internal gases results in the formation of complex plasma systems. Consequently, novel statistical methodologies are necessitated to investigate their thermophysical properties. So, in this work, we will re-study Chandrasekhar’s condition for a state if the gas is assumed to follow the three-parameter non-Maxwell distribution, and we analyze the effects of the three-parameter non-Maxwell distribution on Chandrasekhar’s condition.

## 2. The Gas Pressure in the Non-Maxwell Distribution

Without loss of generality, we consider a cloud of single-component gas, ignoring relativistic effects and assuming it in the non-Maxwellian sense with the distribution function in Equation (6). According to the kinetic theory, the gas pressure is defined as(15)$$P_{g}=\frac{1}{3}mn\;<v^{2}>$$where *n* is the particle number density, *m* is the particle mass, and <*v*^2^> is the mean square velocity of the particle, with(16)$$\langle v^{2}\rangle =\frac{\int v^{2}f\left(v\right)d^{3}v}{\int f\left(v\right)d^{3}v}.$$We substitute the distribution function Equation (6) into Equation (16), and then, the mean square velocity can be derived as(17)$${\langle v^{2}\rangle }_{r,q,\alpha }=\frac{\int_{0}^{\infty }v^{2}\left(1+\alpha \frac{v^{4}}{v_{T}^{4}}\right){\left[1+\frac{1}{q-1}{\left(\frac{v^{2}}{X_{r,q}v_{T}^{2}}\right)}^{r+1}\right]}^{-q}4\pi v^{2}dv}{\int_{0}^{\infty }\left(1+\alpha \frac{v^{4}}{v_{T}^{4}}\right){\left[1+\frac{1}{q-1}{\left(\frac{v^{2}}{X_{r,q}v_{T}^{2}}\right)}^{r+1}\right]}^{-q}4\pi v^{2}dv}\;=\frac{A^{\frac{-1}{r+1}}\int_{0}^{\infty }\frac{x^{\frac{5}{2r+2}-1}}{{\left(1+x\right)}^{q}}dx+\frac{\alpha }{v_{T}^{4}}A^{\frac{-4}{r+1}}\int_{0}^{\infty }\frac{x^{\frac{9}{2r+2}-1}}{{\left(1+x\right)}^{q}}dx}{\int_{0}^{\infty }\frac{x^{\frac{3}{2r+2}-1}}{{\left(1+x\right)}^{q}}dx+\frac{\alpha }{v_{T}^{4}}A^{\frac{-3}{r+1}}\int_{0}^{\infty }\frac{x^{\frac{7}{2r+2}-1}}{{\left(1+x\right)}^{q}}dx}.$$Here, we have let replacements $x=Av^{2(r+1)}$ and $A=1/\left[\left(q-1\right){\left(X_{r,q}{v_{T}}^{2}\right)}^{r+1}\right]$. Using the Euler beta function, when *m* > 0 and *n* > 0, we have that(18)$$B\left(m,n\right)=\int_{0}^{\infty }\frac{x^{m-1}}{{\left(1+x\right)}^{m+n}}dx=\frac{\Gamma \left(m\right)\Gamma \left(n\right)}{\Gamma \left(m+n\right)}.$$So, when *q* > 1 and *q*(*r* + 1) > 9/2, we have that(19)$${\langle v^{2}\rangle }_{r,q,\alpha }=\frac{A^{\frac{-1}{r+1}}\left[B\left(\frac{5}{2r+2},q-\frac{5}{2r+2}\right)+\frac{\alpha }{v_{T}^{4}}A^{\frac{-2}{r+1}}B\left(\frac{9}{2r+2},q-\frac{9}{2r+2}\right)\right]}{B\left(\frac{3}{2r+2},q-\frac{3}{2r+2}\right)+\frac{\alpha }{v_{T}^{4}}A^{\frac{-2}{r+1}}B\left(\frac{7}{2r+2},q-\frac{7}{2r+2}\right)},$$
which equals(20)$${\langle v^{2}\rangle }_{r,q,\alpha }=3Z_{r,q,\alpha }v_{T}^{2},$$
with the modification factor(21)$$Z_{r,q,\alpha }=\frac{G_{q}^{3}\left(\frac{5}{2r+2}\right)+9\alpha G_{q}^{3}\left(\frac{3}{2r+2}\right)G_{q}\left(\frac{9}{2r+2}\right)}{G_{q}^{3}\left(\frac{5}{2r+2}\right)+9\alpha G_{q}\left(\frac{3}{2r+2}\right)G_{q}\left(\frac{5}{2r+2}\right)G_{q}\left(\frac{7}{2r+2}\right)},$$
where we have used the abbreviation $G_{q}(x)=\Gamma \left(x\right)\Gamma \left(q-x\right)$. Therefore, we can write the gas pressure in the three-parameter non-Maxwell distribution as(22)$$P_{g}=\frac{1}{3}mn{\langle v^{2}\rangle }_{r,q,\alpha }=Z_{r,q,\alpha }\frac{k}{\mu m_{H}}\rho T,$$
where, as usual in astrophysics, one uses the mean molecular weight *μ* and the mass of a hydrogen atom *m_H_* to replace the particle mass by *m = μm_H_*. It is clear that the gas pressure in the non-Maxwell distribution is significantly different from that in a Maxwellian distribution. From Equation (21), it is easy to verify that when one takes *r* = 0 and *q* = *κ* + 1, one can obtain the modification factor in the Vasyliunas–Cairns distribution:(23)$$Z_{\kappa ,\alpha }=\frac{\left(\kappa -\frac{5}{2}\right)\left(\kappa -\frac{7}{2}\right)+35\alpha {\left(\kappa -\frac{3}{2}\right)}^{2}}{\left(\kappa -\frac{5}{2}\right)\left(\kappa -\frac{7}{2}\right)+15\alpha \left(\kappa -\frac{3}{2}\right)\left(\kappa -\frac{7}{2}\right)}.$$Further, when one takes *κ →* ∞, one can obtain the modification factor in the Cairns distribution:(24)$$Z_{\alpha }=\frac{1+35\alpha }{1+15\alpha }.$$Only if we take *r =* 0, *α* = 0 and *q* → ∞ do the modification factor *Z_r,q,α_* → 1 and then the gas pressure Equation (22) reach the standard gas pressure (the equation of state of an ideal gas) in a Maxwellian velocity distribution. However, when one takes *α* = 0, one can obtain the modification factor in the (*r*,*q*)-distribution to be 1:(25)$$Z_{r,q}=\frac{\Gamma^{3}\left(\frac{5}{2r+2}\right)\Gamma^{3}\left(q-\frac{5}{2r+2}\right)}{\Gamma^{3}\left(\frac{5}{2r+2}\right)\Gamma^{3}\left(q-\frac{5}{2r+2}\right)}\equiv 1\;.$$When one takes *r* = 0, *α* = 0 and *q* = *κ* + 1, one can also obtain the modification factor in the *κ*-distribution to be 1:(26)$$Z_{\kappa }=\frac{\Gamma^{3}\left(\frac{5}{2}\right)\Gamma^{3}\left(\kappa +1-\frac{5}{2}\right)}{\Gamma^{3}\left(\frac{5}{2}\right)\Gamma^{3}\left(\kappa +1-\frac{5}{2}\right)}\equiv 1\;,$$
so that the (*r*,*q*)-distribution and the *κ*-distribution have no effect on the gas pressure.

## 3. Generalized Chandrasekhar Condition of a Gas Star

We now study the Chandrasekhar condition for a star based on the gas pressure (Equation (22)) in the three-parameter non-Maxwellian distribution. We may assume that the new three-parameter distribution only affects the gas pressure, as shown in Equation (22), but ignore the effect on radiation. The contribution of the gas pressure to the total pressure is a fraction *β*, and the radiation pressure *P_r_* = *aT*^4^/3 contributes the other (1 − *β*), where *a* is Stefan’s constant. With the gas pressure in Equation (22), the total pressure can be written as(27)$$P=\frac{1}{\beta }Z_{r,q,\alpha }\frac{k}{\mu m_{H}}\rho T=\frac{1}{1-\beta }\frac{1}{3}aT^{4}.$$
From Equation (27), the temperature can be derived as(28)$$T={\left(Z_{r,q,\alpha }\frac{3}{a}\frac{1-\beta }{\beta }\frac{k}{\mu m_{H}}\rho \right)}^{\frac{1}{3}}.$$By substituting Equation (28) back into Equation (27), we have(29)$$P={\left[\frac{3\left(1-\beta \right)}{a}\right]}^{\frac{1}{3}}{\left(Z_{r,q,\alpha }\frac{1}{\beta }\frac{k}{\mu m_{H}}\rho \right)}^{\frac{4}{3}}.$$Using this pressure from Equation (29), we can write the pressure at the center of a star, and on the basis of the inequality Equation (1), we can obtain the condition for the stable existence of a star in the three-parameter non-Maxwell distribution, namely,(30)$$P_{c}={\left[\frac{3\left(1-\beta_{c}\right)}{a}\right]}^{\frac{1}{3}}{\left(Z_{r,q,\alpha }\frac{1}{\beta_{c}}\frac{k}{\mu m_{H}}\rho_{c}\right)}^{\frac{4}{3}}\leq \frac{1}{2}G{\left(\frac{4\pi }{3}\right)}^{\frac{1}{3}}{\rho_{c}}^{\frac{4}{3}}M^{\frac{2}{3}},$$
where *β_C_* is *β* at the center of the star. With Stefan’s constant $a=8\pi^{5}k^{4}/(15h^{3}c^{3})$, from Equation (30), we derive that(31)$$Z_{r,q,\alpha }^{2}{\left(\frac{135}{4\pi^{6}}\right)}^{\frac{1}{2}}{\left(\frac{hc}{G}\right)}^{\frac{3}{2}}\frac{1}{m_{H}^{2}}{\left(\frac{1-\beta_{c}}{\beta_{c}^{4}}\right)}^{\frac{1}{2}}\leq \mu^{2}M\;.$$To further simplify Equation (31), we take ${\left[135/\left(4{\pi \;}^{6}\right)\right]}^{1/2}\approx 0.1873$ and give the combination of these nature constants in units of the Sun’s mass with ${\left(hc/G\right)}^{\frac{3}{2}}/m_{H}^{2}$ ≈ $29.2M_{⊙}$. Then, we have(32)$$5.48M_{⊙}{\left(\frac{1-\beta_{c}}{{\beta_{c}}^{4}}\right)}^{\frac{1}{2}}Z_{r,q,\alpha }^{2}\leq \mu^{2}M.$$From Equation (32), we can obtain the maximum fraction of the radiation pressure at the center of the star with a given mass *M*. If the limiting fraction of gas pressure with the three-parameter non-Maxwell distribution is now $\beta_{**}$ and the upper limit of the radiation pressure is 1 − $\beta_{**}$, we can obtain the following: (33)$$5.48{\left(\frac{1-\beta_{**}}{\beta_{**}^{4}}\right)}^{\frac{1}{2}}Z_{r,q,\alpha }^{2}=\frac{\mu^{2}M}{M_{⊙}},$$
and thus, the condition in (4) is replaced by (33), and Chandrasekhar’s condition of ideal gas stars is generalized to that of the gas with a three-parameter non-Maxwell distribution, where the modification factor *Z_r_*_,_*_q_*_,*α*_ plays an important role. From Equation (33), we can find the upper limit of the gas and radiation pressure in the gases with various non-Maxwell distributions when we select different values of the three parameters (*r*, *q*, *α*). Only when we take *r* = 0, *α* = 0 and *q* → ∞ do we have *Z_r_*_,_*_q_*_,*α*_ → 1, and Equation (33) reduces to the classical condition for an ideal gas with a Maxwellian distribution (Equation (4)) obtained by Chandrasekhar [4].

## 4. Generalized Chandrasekhar Condition of a Centrally Condensed Star

Now, we consider a centrally condensed star. For the sake of convenience, we simply divide the interior of a star into three zones where the gas pressure, the electron degeneracy pressure, and the relativistic degeneracy pressure fight against gravity. And we consider surfaces with the same pressure in them, as Chandrasekhar did in his work [5]. In this paper, we only discuss the effects of the three-parameter non-Maxwellian distributions in the gas envelope.

In the first zone Z_1_, the total pressure *P* of a gas cloud with the three-parameter non-Maxwell distribution can be obtained by Equation (29). Then, the gas pressure in the gas envelope of the star can be expressed as(34)$$P_{g}=\frac{1}{\beta }P=C_{*}\rho^{\frac{4}{3}},$$
with(35)$$C_{*}={\left[\frac{3\left(1-\beta \right)}{a\beta }\right]}^{\frac{1}{3}}{\left(Z_{r,q,\alpha }\frac{k}{\mu m_{H}}\right)}^{\frac{4}{3}}.$$
In the degenerate zone Z_2_, the equation of state of the gas is given by(36)$$P_{d}=K_{1}\rho^{\frac{5}{3}},$$
with(37)$$K_{1}=\frac{1}{20}{\left(\frac{3}{\pi }\right)}^{\frac{2}{3}}\frac{h^{2}}{m_{e}}{\left(\frac{1}{\mu m_{H}}\right)}^{\frac{5}{3}}\;.$$
At the surface S_1_ between the gas envelope and the degenerate zone, the pressure should be the same, and thus, from Equations (34) and (37), we can find the density *ρ*_1_ at the surface:(38)$${C_{*}\rho_{1}}^{\frac{4}{3}}=K_{1}{\rho_{1}}^{\frac{5}{3}},$$
or(39)$$\rho_{1}={\left(\frac{C_{*}}{K_{1}}\right)}^{3}\;.$$In the central zone Z_3_, the relativistic–degenerate equation of state of the gas is(40)$$P_{rd}=K_{2}\rho^{\frac{4}{3}},$$
with(41)$$K_{2}=\frac{hc}{8}{\left(\frac{3}{\pi }\right)}^{\frac{1}{3}}{\left(\frac{1}{\mu m_{H}}\right)}^{\frac{4}{3}}.$$From Equations (37) and (41), the density *ρ*_2_ at the second surface S_2_ between Z_2_ and Z_3_ is given by(42)$$K_{2}{\rho_{2}}^{\frac{4}{3}}=K_{1}{\rho_{2}}^{\frac{5}{3}},$$
or(43)$$\rho_{2}={\left(\frac{K_{2}}{K_{1}}\right)}^{3}.$$

In centrally condensed stars, there should be $\rho_{2}>\rho_{1}$; then, we have that(44)$${\left(\frac{K_{2}}{K_{1}}\right)}^{3}>{\left(\frac{C_{*}}{K_{1}}\right)}^{3},$$
or(45)$${\left(\frac{hc}{8}\right)}^{3}\frac{3}{\pi }{\left(\frac{1}{\mu m_{H}}\right)}^{4}>\frac{3\left(1-\beta \right)}{a\beta }{\left(Z_{r,q,\alpha }\frac{k}{\mu m_{H}}\right)}^{4}\;.$$
Then, it can be obtained from Equation (45) that(46)$$\frac{1-\beta }{\beta }<\frac{h^{3}c^{3}a}{256\pi k^{4}Z_{r,q,\alpha }^{4}}=0.1015\frac{1}{Z_{r,q,\alpha }^{4}},$$
or(47)$$1-\beta <\frac{0.1015}{0.1015+Z_{r,q,\alpha }^{4}}.$$If the limiting fraction of gas pressure in the centrally condensed star is $\beta_{***}$, and the limit of radiation pressure is 1 − $\beta_{***}$, we have that(48)$$1-\beta_{***}=\frac{0.1015}{0.1015+Z_{r,q,\alpha }^{4}},$$Thus, we obtain the range of the radiation pressure allowed in a centrally condensed star in the three-parameter non-Maxwell distribution, where the modification factor *Z_r_*_,_*_q_*_,*α*_ plays an important role. Only when we take *r* = 0, α = 0, and *q* → ∞ do we have that *Z_r_*_,_*_q_*_,*α*_ → 1, and Equation (47) reduces to the classical condition for a centrally condensed star, 1−β<0.0921 in Equation (5), as obtained by Chandrasekhar [5].

## 5. Numerical Analyses

From the generalized Chandrasekhar condition in Equation (33), we find that the maximum radiation pressure at the center of a stable gas star now depends on the three non-Maxwellian distribution parameters (*r*, *q*, *α*) with the modification factor *Z_r_*_,_*_q_*_,*α*_. To more intuitively show the relationship between the maximum radiation pressure and the mass of a star and its dependence on the three parameters (*r*, *q*, *α*) in the non-Maxwellian distribution, numerical analyses are shown in Figure 1, Figure 2 and Figure 3. The purpose here is to illustrate the property of Chandrasekhar’s condition when the gas deviates from the Maxwellian distribution, so the selection of three parameters in the numerical analyses mainly considers the degree of deviation from the values (*r*, *q*, *α*) = (0, ∞, 0).

In Figure 1, we show the relation between the maximum radiation pressure 1 − $\beta_{**}$ and the mass of the star (if we take the mean molecular weight *μ* as a constant) with four values of the parameter *r*, where the parameters *q* and *α* are fixed at *q* = 50 and *α* = 0.06. When *r =* 0, the three-parameter distribution is equivalent to the Vasyliunas–Cairns distribution, and the maximum radiation pressure is much lower than its value in the Maxwellian sense. As *r* increases, the maximum radiation pressure in the star will gradually increase. When we take *r* → ∞, we have $\underset{r\to \infty }{\lim}Z_{r,q,\alpha }=1$. This means that an increase in *r* will reduce the influence of *q* and *α*, and the maximum radiation pressure will gradually return to the value in a Maxwellian distribution.

In Figure 2, we show the relation between the maximum radiation pressure 1 − $\beta_{**}$ with values of parameter *q*, where the parameters *r* and *α* are fixed at *r =* 3 and *α* = 0.06. As *q* decreases, the maximum radiation pressure in the star generally decreases. However, the curve of *q* → ∞ is almost the same as the curve of *q* = 50, and the maximum radiation pressure will change significantly only when the value of *q* decreases to a very low value, which means it deviates seriously from the Maxwellian distribution.

In Figure 3, we show the relation between the maximum radiation pressure 1 − $\beta_{**}$ with values of parameter *α*, where the parameters *r* and *q* are fixed at *r* = 3 and *q* = 50. It is worth noting that when we take *α* = 0, the three-parameter distribution will reduce to the (*r*, *q*)-distribution, and we will find that *Z_r_*_,_*_q_*_,*α*_ always equals 1 and the result of Equation (33) will be the same as that of Equation (4). This means that if we only consider the kappa distribution or the (*r*, *q*)-distribution, the maximum radiation pressure will always be the same as its value in the Maxwellian sense. This is because one demands of the kappa distribution function that the most probable speed of the particle should be obtained from the second moment of the distribution with a given value of mean particle energy in a self-consistent manner [30]. Then, the most probable speed in the kappa distribution [31] is considered to be $\sqrt{2kT\left(\kappa -3/2\right)/(\kappa m)}$, distinct from the Maxwellian thermal speed $\sqrt{2kT/m}$, and this will result in the second moment of the velocity of the particles which follow the kappa distribution being independent of the kappa parameter. The (*r*, *q*)-distribution also retains this property. However, the three-parameter distribution has a structure similar to the Cairns distribution, in which the second moment of the velocity can be changed by the *α* parameter, so *κ* and *r* can have an effect on Chandrasekhar’s condition when *α* ≠ 0. From Figure 3, we find that when *α* is not equal to zero, the maximum radiation pressure in the star decreases rapidly with an increase in *α* at first, and then slowly approaches a value determined by the parameters *r* and *q*.

In all the analyses, we conclude for a gas star that although all the three parameters can affect the maximum radiation pressure, they do not change the trend of its increase with an increase in star mass. And in general, the non-Maxwellian distribution can make the maximum radiation pressure smaller than its value in the Maxwellian sense.

As for centrally condensed stars, in Equation (47), we also performed some numerical analyses of the maximum radiation pressure in the gas envelope, and we have listed them in Table 1, Table 2 and Table 3. In the first row of these tables, we list the maximum radiation pressure in the Maxwellian sense for centrally condensed stars. Then, we fix two of the three parameters and analyze how the maximum radiation pressure changes with the other parameter.

In Table 1, we can see that when *q* and *α* are fixed at *q* = 50 and *α* = 0.06, the maximum radiation pressure 1 − $\beta_{***}$ in a centrally condensed star gradually increases and eventually approaches the value in the Maxwellian sense.

In Table 2, we find that the effect of the parameter *q* on the maximum radiation pressure in a centrally condensed star is similar to that in an ideal gas star. When *q* is large, its change has little effect on the maximum radiation pressure. Only until it drops to a very small value will the maximum radiation pressure experience a significant reduction.

In Table 3, when we fix *q* = 50 and *r* = 3, we can also see that the parameter *α* has a similar effect in a centrally condensed star as it does in an ideal gas star. As *α* gradually increases, the maximum radiation pressure in the star will rapidly decrease and then approach a value determined by the parameters *q* and *r*.

In combination with the above analysis, we conclude that the three-parameter non-Maxwellian distribution acts very similarly in centrally condensed stars as they do in an ideal gas star. In general, the non-Maxwell distribution will also make the maximum radiation pressure in the gas envelope of a centrally condensed star smaller than the result calculated with a Maxwellian distribution.

## 6. Conclusions and Discussion

In conclusion, we have re-studied Chandrasekhar’s conditions for the equilibrium and stability of stars and analyzed the limit of the radiation pressure in a gas star and a centrally condensed star with a universal three-parameter non-Maxwell distribution, which contains extensive non-Maxwell velocity distributions. We have derived expressions of the maximum radiation pressure with the three-parameter distribution for a gaseous star and a centrally condensed star and thus generalized Chandrasekhar’s conditions for stars.

By some specific numerical analyses with different values of the non-Maxwell parameters, we conclude that the non-Maxwellian distributions have significant effects on Chandrasekhar’s conditions. They usually reduce the maximum radiation pressure prediction in both gaseous and centrally condensed stars. Because the non-Maxwellian distribution changes the state equation of a star and thus modifies Chandrasekhar’s conditions for the equilibrium and stability of stars with a Maxwellian distribution, the results may impact predictions of the internal structure of stars, such as the limiting proportions of radiation zones and convection zones within stable stars, as well as energy transport efficiency in them and even the mass of a star based on Equation (4) if the gases are not in a thermal equilibrium state with a Maxwellian distribution, which helps us to gain a more comprehensive understanding of stellar structures and evolutions if the particles in astrophysical systems have non-Maxwellian velocity distributions.

This manuscript presents a generalization of Chandrasekhar’s condition in a universal three-parameter non-Maxwell distribution that contains all known non-Maxwellian distributions (as well as the Maxwellian distribution) when one takes different selections of three parameters (*r*, *α*, *q*). In future work, if we try to apply the generalized Chandrasekhar condition of this work to study the stability of a specific star with one of the non-Maxwell velocity distributions, we might indirectly determine the values of the parameters in the velocity distribution function based on the observational evidence of the specific problem. Employing helioseismology or asteroseismology to indirectly infer the parameters in the non-Maxwellian velocity distribution within a stellar interior may be a promising methodological pathway.

## Figures and Tables

**Figure 1 entropy-27-00470-f001:**
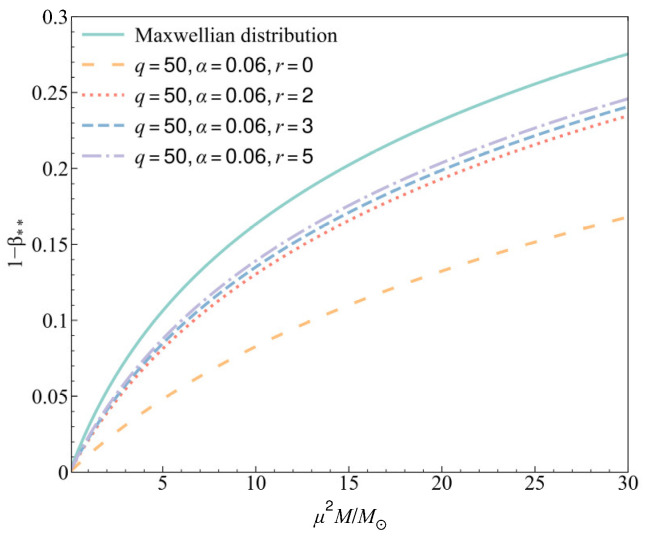
The maximum radiation pressure in a gas star for different values of *r* with *q* = 50 and *α* = 0.06.

**Figure 2 entropy-27-00470-f002:**
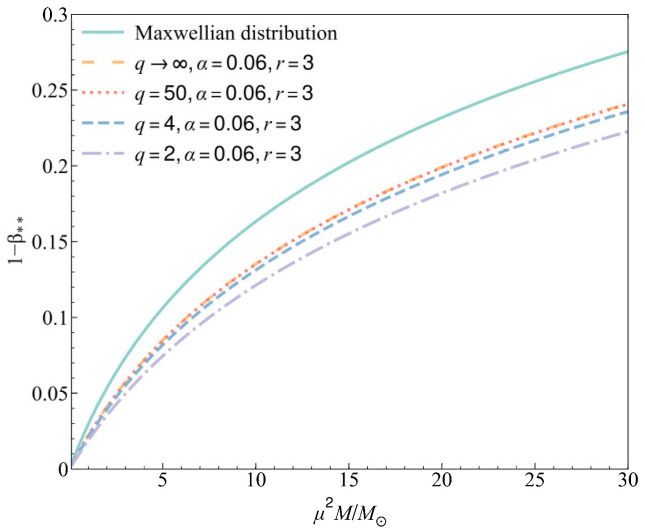
The maximum radiation pressure in a gas star for different values of *q* with *r* = 3 and *α* = 0.06.

**Figure 3 entropy-27-00470-f003:**
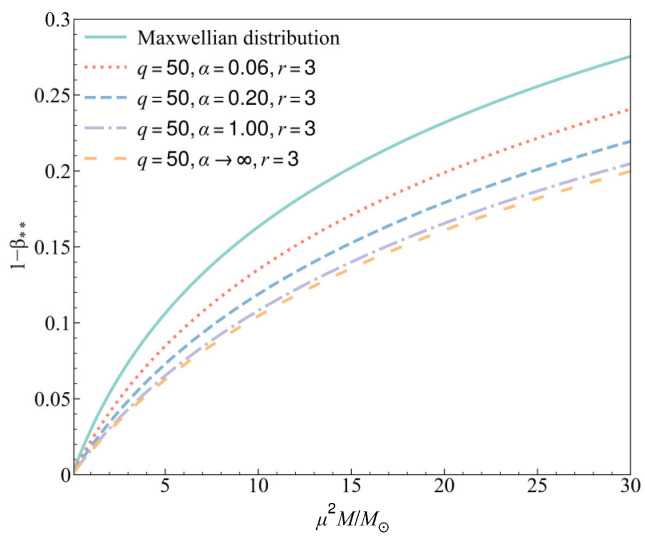
The maximum radiation pressure in ideal gas stars for different values of *α* with *r* = 3 and *q* = 50.

**Table 1 entropy-27-00470-t001:** The maximum radiation pressure in centrally condensed stars for different values of *r* with *q* = 50 and *α* = 0.06.

*r*	*q*	*α*	*Z_r,q,α_*	$1-\beta_{***}$
0	∞	0	1.00	9.21%
0	50	0.06	1.69	1.24%
2	50	0.06	1.21	4.54%
3	50	0.06	1.17	5.06%
5	50	0.06	1.15	5.55%
∞	50	0.06	1	9.21%

**Table 2 entropy-27-00470-t002:** The maximum radiation pressure in centrally condensed stars for different values of *q* with *r* = 3 and *α* = 0.06.

*r*	*Q*	*α*	*Z_r,q,α_*	$1-\beta_{***}$
0	∞	0	1.00	9.21%
3	∞	0.06	1.17	5.09%
3	50	0.06	1.17	5.06%
3	4	0.06	1.20	4.63%
3	2	0.06	1.28	3.64%

**Table 3 entropy-27-00470-t003:** The maximum radiation pressure in centrally condensed stars for different values of *α* with *q* = 50 and *r* = 3.

*r*	*q*	*α*	*Z_r,q,α_*	$1-\beta_{***}$
0	∞	0	1.00	9.21%
3	50	0.06	1.17	5.06%
3	50	0.20	1.30	3.44%
3	50	1.00	1.40	2.60%
3	50	∞	1.43	2.36%

## Data Availability

Data are contained within the article.

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
