# Peer review of "Chandrasekhar’s Conditions for the Equilibrium and Stability of Stars in a Universal Three-Parameter Non-Maxwell Distribution"

_entropy, 2025, doi:10.3390/e27050470_

Round 1
Reviewer 1 Report
Comments and Suggestions for Authors
Under a universal three-parameter non-Maxwell distribution, in this manuscript the authors address the Chandrasekhar's conditions for equilibrium and stability of stars. The paper presents interesting results with possible applications to white dwarfs, and the mathematical calculations appear to be correct. I think it could be considered for publication; however, there are some solid clarifications and physical motivations to be taken into account in the preparation of a revised version. My queries/suggestions are listed below:
- Here I will make some general criticisms about the writing of the paper: All equations should be numbered, and many of them in the manuscript are not. After Eq. (13), the first word should begin with a capital letter. Furthermore, figures and tables are not properly referenced, making it difficult to identify the specific figure being discussed in the text (resulting in the message "Error! Reference source not found"). The authors should verify all these observations carefully throughout the text.
- It is not clear why a star (such as a white dwarf) would have non-Maxwell velocity distributions? The authors state: "The recent observations of complex systems in space and astrophysical plasmas may indicate the universal existence of non-Maxwell distributions in the interior of stars", but on what basis? What are the references for such recent observations?
- When the authors discuss "Equilibrium and Stability of Stars" what kind of stability are they referring to? This should be made clear because we often refer to the stability of equilibrium stars when they are radially perturbed; see, for example, the classic papers by Chandrasekhar [ApJ 140, 417 (1964) and PRL 12, 114 (1964)].
- On line 177, page 6, β** should be correctly written as $\beta_{**}$. This should also be corrected in the last column of tables 1, 2, and 3. Furthermore, it is necessary to specify what the two asterisks (**) in Eq. (28) mean instead of just one asterisk as in (4).
- In all figures, random values ​​have been assumed for the three non-Maxwellian distribution parameters (\alpha, r, q), but there is no clear and reasonable justification for choosing such specific values. It would be quite fruitful and novel to use values ​​of (\alpha, r, q) that can describe observed white dwarfs in the universe. In other words, given recent astronomical observations, is it possible to constraint the parameter space (\alpha, r, q) in this manuscript?
- Almost 30% of the references correspond to one of the authors, making the paper highly self-citable. It would be pertinent to also consider relevant references from other authors in order to better standardize the bibliography.
I will be available to receive the revised version and reserve my decision until my comments are clearly and precisely answered/addressed.
Author Response
Comment1: Here I will make some general criticisms about the writing of the paper: All equations should be numbered, and many of them in the manuscript are not. After Eq. (13), the first word should begin with a capital letter. Furthermore, figures and tables are not properly referenced, making it difficult to identify the specific figure being discussed in the text (resulting in the message "Error! Reference source not found"). The authors should verify all these observations carefully throughout the text.
Response1: All equations in the manuscript have now been properly numbered. The letter following mentioned equation has been uppercase. References to figures and tables have also been corrected.
Comment2: It is not clear why a star (such as a white dwarf) would have non-Maxwell velocity distributions? The authors state: "The recent observations of complex systems in space and astrophysical plasmas may indicate the universal existence of non-Maxwell distributions in the interior of stars", but on what basis? What are the references for such recent observations
Response2: In page 4, we have added the explanations on this point and the corresponding references.
Comment3: When the authors discuss "Equilibrium and Stability of Stars" what kind of stability are they referring to? This
should be made clear because we often refer to the stability of equilibrium stars when they are radially perturbed; see, for example, the classic papers by Chandrasekhar [ApJ 140, 417 (1964) and PRL 12, 114 (1964)].
Response3: As a generalization of Chandrasekhar’s condition, the equilibrium and stability of stars in this paper is the same as that described in the work by Chandrasekhar in MNRAS 96, 644 (1936) and Rev. Mod. Phys. 56, 137 (1984), as equivalent to the condition for the stable existence of stars[3,4].
Comment4: On line 177, page 6, β** should be correctly written as $\beta_{**}$. This should also be corrected in the last column of tables 1, 2, and 3. Furthermore, it is necessary to specify what the two asterisks (**) in Eq. (28) mean instead of just one asterisk as in (4).
Response4: The points have been corrected and explained in the revised manuscript.
Comment5: In all figures, random values have been assumed for the three non-Maxwellian distribution parameters (\alpha, r,
q), but there is no clear and reasonable justification for choosing such specific values. It would be quite fruitful and novel to use values of (\alpha, r, q) that can describe observed white dwarfs in the universe. In other words, given recent astronomical observations, is it possible to constraint the parameter space (\alpha, r, q) in this manuscript?
Response5: For this point, at the bottom in page 8 of the revised manuscript, we have added an explanation on the selection of the three parameters in the numerical analyses.
Comment6: Almost 30% of the references correspond to one of the authors, making the paper highly self-citable. It would be
pertinent to also consider relevant references from other authors in order to better standardize the bibliography.
Response6: In the revised manuscript, we have made adjustments for the cited references.
Reviewer 2 Report
Comments and Suggestions for Authors
In the manuscript the authors re-investigate the condition for stellar equilibrium in the presence of radiation, initially considered by Chandrasekhar, by assuming that the stellar material is described by the non-Maxwellian distribution (6), which leads to the expression (17) of the gas pressure. Then the Chandrasekhar equilibrium condition is generalized in Eq. (28), and the case of degenerate matter is also considered.
The manuscript may be publishable in Symmetry if the authors would fully consider the following points:
- There is a problem with the presentation of the Figures and Tables, and their labelling, which must be corrected.
- What is the physical and mathematical justification of the distribution (6), except the fact that it unifies a number of known distributions?
- The authors should also discuss in some detail the astrophysical implications of their results, and how they can be useful for a better understanding of the stellar structures. For example, what would the effect of the modified polytropic equation of state (29) on the mass of the star?
- Do the modified equilibrium conditions have an impact on the structure or stability of the Sun? Would it be possible to test these modifications in the Maxwellian distribution observationally?
e
Comments on the Quality of English LanguageQuality of English language generally fine.
Author Response
Comment1: There is a problem with the presentation of the Figures and Tables, and their labelling, which must be corrected.
Response1: The formatting issues in figures and tables have been corrected in the revised manuscript.
Comment2: What is the physical and mathematical justification of the distribution (6), except the fact that it unifies a number of known distributions?
Response2: The three-parameter non-Maxwellian distribution function (6) has served as a unified framework that contains all the known non-Maxwellian distributions (as well as the Maxwellian distribution) when one takes different values of the three parameters (r, α, q). We think, its importance is just that at present.
Comment3: The authors should also discuss in some detail the astrophysical implications of their results, and how they can
be useful for a better understanding of the stellar structures. For example, what would the effect of the modified polytropic equation of state (29) on the mass of the star?
Response3: In Sec.6 of the revised manuscript, we have added discussion on the astrophysical implications of the results.
Comment4: Do the modified equilibrium conditions have an impact on the structure or stability of the Sun? Would it be
possible to test these modifications in the Maxwellian distribution observationally?
Response4: Yes, we think. Because the non-Maxwellian distribution changes the state equation of a star and thus modified the Chandrasekhar’s conditions as equilibrium and stability of stars with a Maxwellian distribution, it may have an impact on the structure or stability of the Sun. The observations of the solar sound speeds in helioseismology have revealed that the velocity
q-distribution in nonextensive statistics, a non-Maxwellian distribution, exists in the interior of the Sun [23] (see Europhys. Lett. 75 (2006) 861-867 ).
Round 2
Reviewer 1 Report
Comments and Suggestions for Authors
The authors have improved their manuscript by taking into account my suggestions and correctly answering most of my questions. Nevertheless, my question 5 ("Given recent astronomical observations, is it possible to constraint the parameter space (\alpha, r, q) in this manuscript?") has not been answered. I had hoped the authors would go beyond mathematical calculations, comparing their theoretical predictions with recent observational evidence. I understand their desire to address Chandrasekhar's condition when the stellar gas deviates from the Maxwellian distribution, but choosing random values for any parameter space ​​without a clear justification (or comparison with observational measurements) weakens the scientific rigor of an article.
However, the manuscript already presents generalizations from previous studies and paves the way for future research, so I suggest its publication in its current form.
Author Response
Comment: The authors have improved their manuscript by taking into account my suggestions and correctly answering most of my questions. Nevertheless, my question 5 ("Given recent astronomical observations, is it possible to constraint the parameter space (\alpha, r, q) in this manuscript?") has not been answered. I had hoped the authors would go beyond mathematical calculations, comparing their theoretical predictions with recent observational evidence. I understand their desire to address Chandrasekhar's condition when the stellar gas deviates from the Maxwellian distribution, but choosing random values for any parameter space ​​without a clear justification (or comparison with observational measurements) weakens the scientific rigor of an article.
Response: For this point, in page 12 of Sec.6 in the revised manuscript we have added a further explanation.